# A randomized clinical trial of a new anti–cervical stenosis device after conization by loop electrosurgical excision

**Marcelo de Andrade Vieira**[1☯]*, **Raphael Leonardo Cunha de Araújo**[2‡], **Carlos Eduardo Mattos da Cunha Andrade**[1‡], **Ronaldo Luis Schmidt**[1‡], **Agnaldo Lopes Filho**[3‡], **Ricardo dos Reis**[1☯]

**1** Department of Gynecologic Oncology, Barretos Cancer Hospital, Barretos, São Paulo, Brazil, **2** Department of Digestive Surgery, Escola Paulista de Medicina, Universidade Federal de São Paulo (UNIFESP), São Paulo, Brazil, **3** Department of Obstetrics and Gynecology, Universidade Federal de Minas Gerais (UFMG), Belo Horizonte, Minas Gerais, Brazil

☯ These authors contributed equally to this work.
‡ These authors also contributed equally to this work.
* mvieiraonco@gmail.com

**Data Availability Statement:** All relevant data are within the manuscript and its Supporting Information files.

## Abstract

### Background

The complications inherent to conization include vaginal bleeding, cervical stenosis, amenorrhea, dysmenorrhea, and deep dyspareunia. Cervical stenosis is the most important complication due to the clinical repercussions. Studies show rates of cervical stenosis ranging from 1.3 to 19% after the Loop Electrosurgical Excision Procedure (LEEP).

### Objective

Our primary outcome was to compare the role of a new endocervical device to prevent cervical stenosis after LEEP in patients with high-grade squamous intraepithelial lesions (HSILs).

### Methods

A randomized clinical trial was performed including phases II and III for evaluation of a new device for cervical stenosis prevention. In Phase II, we included 25 patients who underwent LEEP and placement of the device to assess its toxicity and efficacy. In phase III, we compared two groups (with and without the use of an anti-stenosis device) to evaluate its efficacy and safety.

### Results

From August 2015 to June 2018, 265 participants were randomized (Phase II: 25, Phase III: 120 with DUDA and 120 without DUDA). The toxicity during phase II was observed in only one patient (4%) with pain grade > 7. There were 7 cases of toxicity during Phase III, 2 in the DUDA group (1.8%), and 5 in the No DUDA group (4.5%). The complications rate was numerically higher in the No DUDA group (2.5x higher) than the DUDA group, but this

**Funding:** Hospital de Amor provided material and financial support for the study, but had no role in study design, data collection and analysis, decision to publish, or preparation of the manuscript. The authors did not receive any additional funding for this work.

**Competing interests:** NO authors have competing interests

difference did not reach statistical significance (p = 0.52). The rate of cervical stenosis in DUDA group was (4–7,3%), and in No DUDA group was (4.3–5.8%) (p = 0.5). We did not find a significant difference when comparing the evolution at 3, 6, and 12 months in terms of cervical patency and visualization of the squamocolumnar junction (SCJ) during colposcopy. The DUDA group exhibited 15% to 19% nonvisualization of the SCJ, whereas that rate ranged from 10 to 12% in the No DUDA group.

## Conclusions

The rate of cervical stenosis was not different comparing the use of a new device, specifically produced to prevent cervical stenosis, compared to no use after LEEP procedure. This clinical trial opens up space for a discussion of the utility of using cervical stenosis devices after LEEP. Perhaps in another type of conization it can be evaluated to avoid cervical stenosis.

## Introduction

Infection with the human papillomavirus (HPV) virus is common among women. In most cases, after 6 to 12 months of infection, the immune system eliminates the virus. If the lesion persists, women may develop cancer precursor lesions in the cervix and, if not treated, cervical cancer [1–4]. The term used for cervical precursor lesions or high-grade cervical intraepithelial neoplasia (CIN) is used to define lesions previously reported as moderate dysplasia (CIN II) and severe dysplasia/carcinoma in situ (CIN III) [5]. The natural history of CIN II indicates that 32% to 43% of untreated lesions will regress without treatment, while 35% to 56% will persist and 14% to 22% will progress to carcinoma in situ or invasive carcinoma [5–7]. When assessing CIN III lesions, studies show that within 30 years after infection, 31% of patients will develop cervical cancer [2]. When we group the premalignant lesions into CIN II and III, studies show that in 10 to 20 years, the progression to cancer will occur in some cases if they are left.

Untreated [8]. Progression depends on factors such as the health system, socioeconomic status, HPV status, smoking and awareness of the importance of Pap smears [9].

Treatment of high-grade CIN (CIN II and III) is chosen according to the patient's colposcopic assessment. In cases of satisfactory colposcopy (visible squamocolumnar junction), both ablative and excisional methods are adequate, whereas in cases of the nonvisible squamocolumnar junction, only excisional treatment is accepted [10, 11]. The complications inherent to conization include vaginal bleeding, cervical stenosis, amenorrhea, dysmenorrhea and deep dyspareunia [12, 13]. Cervical stenosis is the most important complication due to the clinical repercussions, which may range from menstrual cramps to hematometra, infertility, and the impossibility of early detection of relapse and or recurrence of the premalignant lesion due to the difficulty of follow-up [12, 14–17]. Studies show rates of cervical stenosis ranging from 3 to 25% after laser conization and from 1.3 to 19% after LEEP [12, 14, 18–21]. Also besides, the diagnosis can be made up to 28 months after conization [14]. At present, there is no consensus standard procedure for the prevention and treatment of cervical stenosis after conization. The clinical studies evaluating anti–cervical stenosis measures have all been nonrandomized, and they have used temporary devices in an attempt to avoid this type of complication [22, 23]. In addition, studies published to date have used several different types of devices without adequate standardization [17, 24, 25]. We propose the first randomized, prospective clinical trial

 

evaluating the role of a new endocervical device called DUDA (Uterine Device to Dilate the Endocervical Canal) developed with characteristics proportional to the size of the cervical and endocervical canal to prevent cervical stenosis after LEEP in patients with high-grade squamous intraepithelial lesions (HSILs).

## Materials and methods

The study was conducted at the Barretos Cancer Hospital from August 2015 to June 2018 (first randomization and the last study follow up). It was started after approval by the Research Ethics Committee (CEP–Local IRB) from Barretos and National Committee in Research Ethics (CONEP–National Committee) because it was on a new medical device (CAAE 36018714.4.0000.5437). It is registered at ClinicalTrials.gov (NCT02500966). After checking the inclusion criteria in the study, the patients were informed about the study and, in accordance with the research terms, signed the Free and Informed Consent Form (ICF). After this step, online randomization was carried out on the website. A randomization list was generated randomly using the R. statistical software. This list was a sequence with dichotomous values divided into even-sized blocks to ensure a balance between patients who would or would not use the DUDA device. Subsequently, this list was attached to a project made on the RedCap Platform to guarantee the accuracy and reliability of data and results [26]. This randomization was blinded, but due to the nature of the study, the patients and physicians were not blinded to the intervention after the outcome of the randomization. The gynecologist who performed the colposcopy during the follow up was blinded to the groups. Furthermore, the patients were allocated to groups with (DUDA Group) or without placement of the DUDA device (No DUDA Group). It was a prospective and randomized manner 1:1. The sample estimate ranged from 104 to 203 in each group. The sample size calculation assumed a rate of cervical stenosis of approximately 15% to 20% after LEEP, as described in the literature, and a reduction of this rate to 6% with the use of an endocervical device, according to the uncontrolled trial of Luesley et al. [22]. Considering this difference in stenosis rates (15–20% vs. 6%), an alpha error of 5%, a beta error of 20%, and balanced allocation of 1:1. All patients signed an informed consent form before joining the study. Patients aged 18 to 65 years with a histopathological diagnosis of CIN2/3 in cervical biopsy and indication for LEEP were included, whereas women were pregnant before LEEP, women with antecedent of LEEP, women who did not understand or did not agree to participate in the study, and women under conditions of high vulnerability (e.g., prisoners, indigenous, etc.) were excluded.

### Study design

We performed phases II and III on the new cervical device according to the standards of randomized clinical trials. In stage II, we included 25 patients who underwent conization and placement of the device to assess its toxicity. In this stage, all patients underwent conization and placement of the device.

In phase III (evaluation of safety and efficacy), the patients were allocated into two groups: 120 patients in the DUDA group, who underwent conization and placement of DUDA, and 120 patients in the No DUDA group who underwent conization but without placement of DUDA.

### Surgical procedure

After intravenous anesthesia, the patients in the lithotomy position received local anesthesia in the cervix at four time-points (3, 6, 9, and 12 hours) with 2% xylocaine without vasoconstrictor using 2 ml per point. Then the LEEP was performed, hemostasis was performed with a roller-

 

ball electrode, the cervix was dilated to Hegar number 8, and DUDA was placed (Figs 1 and 2) on the cervix and secured with four Prolene 2–0 stitches in the DUDA group. Patients in the No DUDA group underwent the LEEP procedure and hemostasis without endocervical canal dilation. Patients were observed for 3 hours and then discharged.

## Follow-up

The patients in phase 1 and phase 3 in the DUDA group removed the DUDA when they returned for a follow-up visit on day 30. The follow-ups were done in phase II at 15 and 30 days and 3, 6, and 12 months. In phase III, they were done at 30 days and 3, 6, and 12 months. The following-up, the squamocolumnar junction (SCJ) and the transformation zone (TZ) were assessed by colposcopy. The gynecologist who performed the colposcopy was blinded to the groups. Both were classified according to international colposcopy standards [27]. During the physical examination, the external visualization of the external cervical orifice (visible or nonvisible/stenosis), menstrual flow (days), menstrual cramps after surgery (mild: spontane-ous improvement; moderate: improvement after the use of oral medications; and severe: requiring intravenous analgesia) and patency of the endocervical canal were assessed through inserting the hysterometer through the canal, classifying the resistance to insertion as none, lit-tle or moderate, or failed insertion (complete resistance to the insertion of the hysterometer). Moreover, oncotic cytology was also collected in all follow-ups. The data were stored on the RedCap platform (Research Electronic Data Capture) [26].

## Characteristics of the device

DUDA consists of a thermoplastic device of the polyoxymethylene (polyacetal). It was designed by the researcher responsible for the study, made by hand in a lathe by machining a cylindrical piece which it was vacuum-packed and sterilized in hydrogen peroxide (Figs 1 and 2). DUDA is cylindrical at one of its ends, with four holes (where the stitches are anchored), is 2.5 cm in diameter and 2 cm long, accompanied by a cylindrical rod with a 4-mm diameter and a central lumen. The rod is introduced into the endocervical canal, and the more substan-tial cylindrical portion makes contact with the external surface of the uterine cervix (Fig 2). DUDA was designed and constructed to adjust to the anatomy and functionality of the uterine cervix in terms of length, thickness, the diameter of the fenestration and base of fixation in the uterine cervix (Fig 2). The device remains in the endocervical canal for 30 days and causes a mechanical obstruction to central healing in an attempt to prevent its stenosis.

## Local tolerance assessment

The adverse events were related to the procedure during the first 30 days after surgery, and the main safety events considered in this analysis were vaginal bleeding and uterine infection which were recorded according to the criteria of CTCAE version 4.0 [28]. If there were a single adverse event grade 3 or higher, the device would be considered unsafe, and consequently, the trial would be discontinued. After evaluation of the first 25 cases by an independent data mon-itoring committee and after analysis and approval of the partial (safety) report by the CEP/CONEP System, the study moved into phase III.

## Objectives

The primary objective of the present study was to evaluate the efficacy of a new cervical device called DUDA in preventing late stenosis of the cervical canal in women undergoing LEEP coniz-ization. The secondary objectives were to evaluate the safety of the device, to compare the SCJ

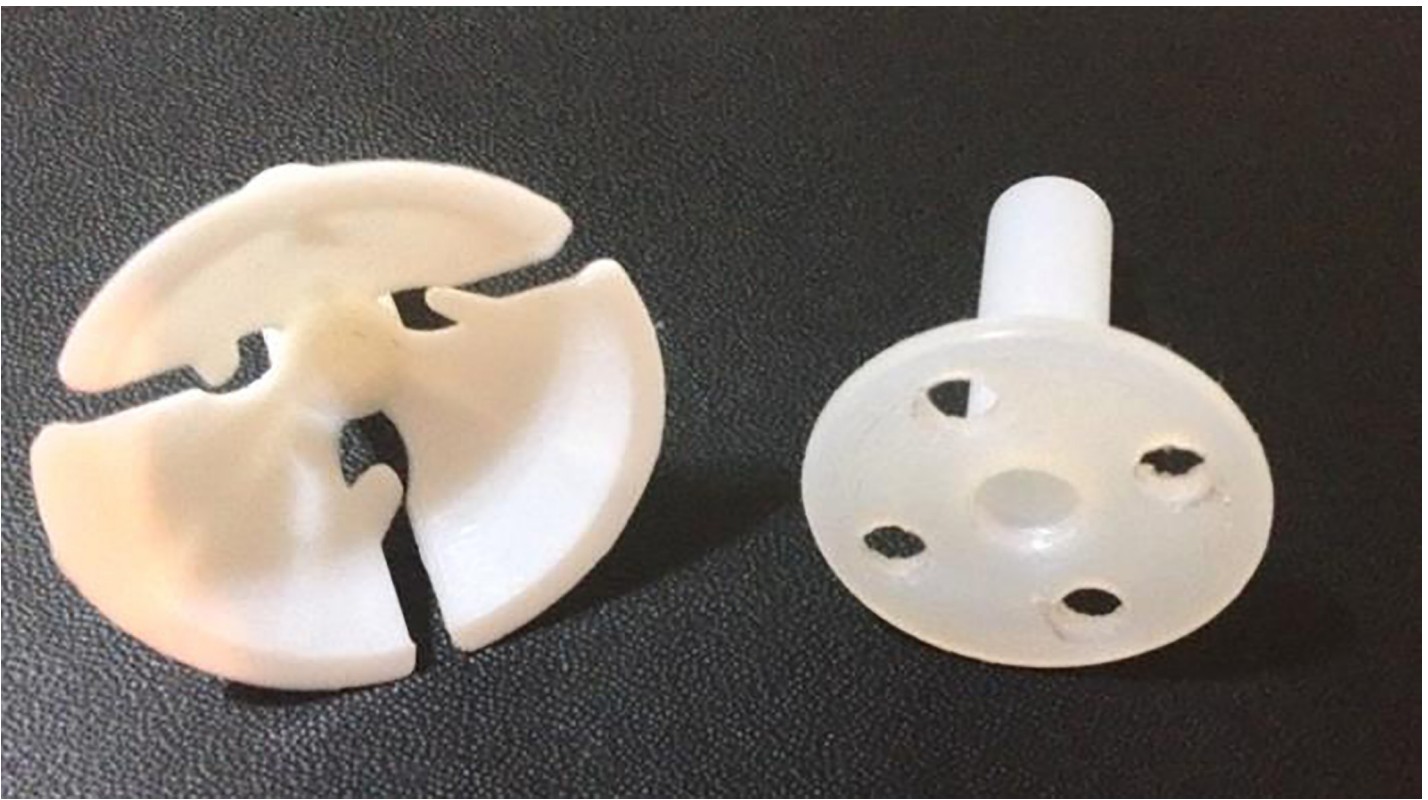

**Fig 1. The balloon holder on the left, the DUDA device on the right.** The DUDA device was created based on the balloon holder.

visualization rate and the rate of patients classified in each type of transformation zone (ZT1, ZT2, and ZT3) between women who did and did not use the device.

## Statistical analysis

Statistical analysis was performed using the chi-squared test or Fisher's exact test. Normality was assessed using the Kolmogorov-Smirnov normality test, and when necessary, a parametric test (t-test), with a description of the mean (m) and standard deviation (SD), and nonparametric test (Mann-Whitney), with a description of the median (md) and interquartile range (IQR), were used. To calculate the sample size according to the difference between the groups for the proportions of stenosis, we used the study by Luesley et al. [26]. To calculate the power of the test according to the sample size and the difference observed between proportions, we used the software GPower v.3.1.9.2. Fisher's exact test was used to calculate the significance of differences in proportions between two independent groups, assuming significance level of 0.05 and a test power of 0.80. For the symptoms of menstruation and dysmenorrhea, we excluded the menopausal women from the analysis.

## Results

### Baseline covariates

A total of 265 patients were selected to participate in the Phase III of the study. However, 20 patients were excluded from randomization and did not enter the study (18 not meeting

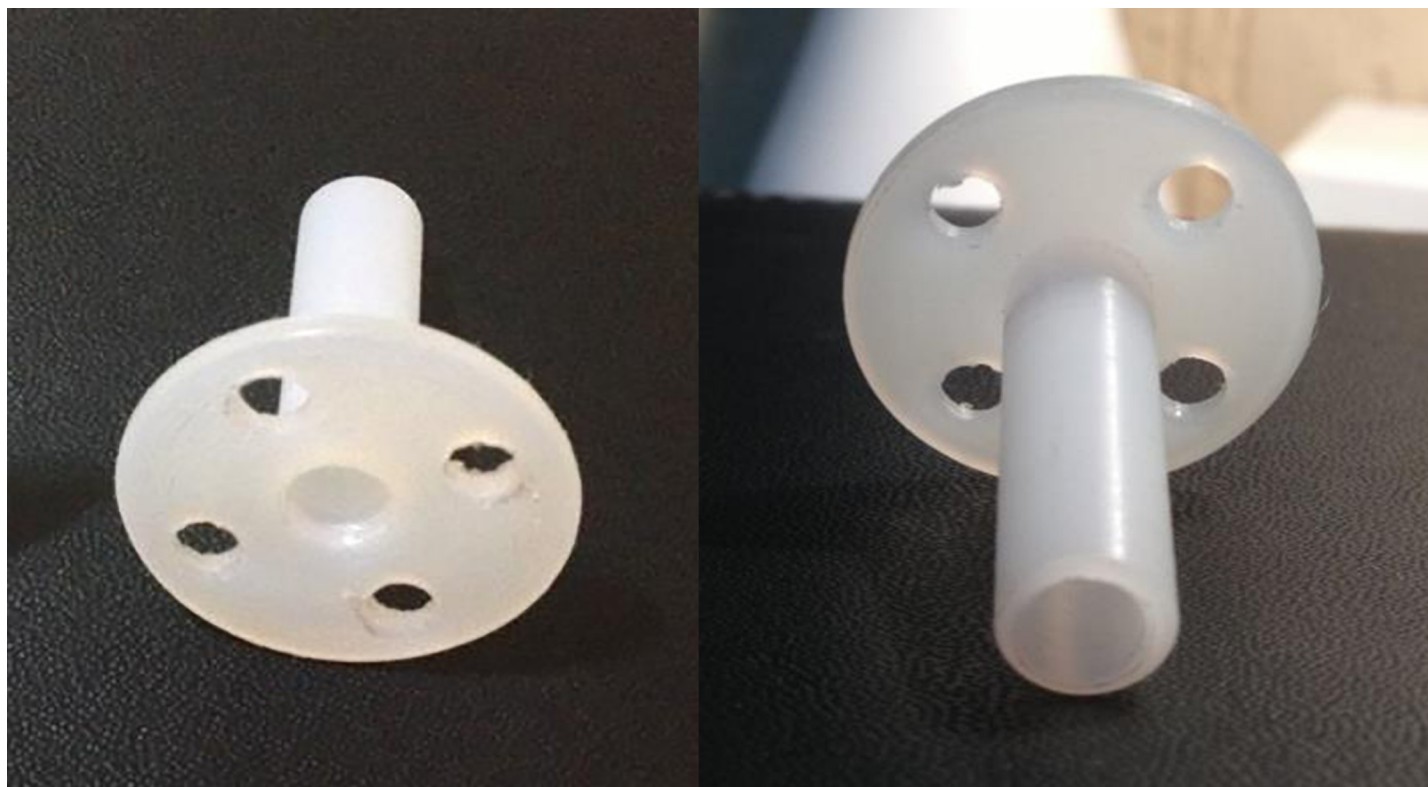

**Fig 2. Device DUDA.** Anterior and posterior views of DUDA.

inclusion criteria and 02 declined to participate in the study) as noted in the CONSORT attachment. All samples underwent LEEP conization.

Patients in phase II had a median age of 34 years old (22–68), had on average 2 children (0–6), were mostly white (60%) and crossbred (32%), and mostly had HSIL (52%) in the initial Pap smear and cervical biopsy with CIN III (44%).

The average duration of placement of the device was 17.7 min (10–30, SD = 6.02 min) (Table 1). All devices were removed within 30 days after the procedure. All patients underwent general and intravenous anesthesia.

**Table 1. Demographic characteristics—phases II and III.**

| | | Phase II | Phase III | Phase III | p-value[*] |
|---|---|---|---|---|---|
| | | N = 25 | DUDA | No DUDA | |
| | | | N = 111 | N = 110 | |
| Duration of Surgery | Median (minutes) (min-max)[a] | 17 (10–30) | | | |
| | Median (minutes) (IQR)[b] | | 15 (6) | 10 (5) | |
| Pregnancy N (%) | | 6 (24) | 1 (0.9) | 3 (2.7) | 0.36 |

[*] p-value comparisons across treatment groups for categorical variables based on Fisher's exact test. P-values for quantitative variables based on Wilcoxon rank-sum test. Normality was assessed using the Kolmogorov-Smirnov normality test, and when necessary, a parametric test (t-test), with a description of the mean (m) and standard deviation (SD), and nonparametric test (Mann-Whitney), with a description of the median (md) and inter-quartile range (IQR), were used.

a- Median (min = minimum, max = maximum)

b Median (inter-quartile range).

**Table 2. Evaluation of the toxicity of the antistenosis device.**

| | Phase II | Phase III | Phase III | |
| | | DUDA | No DUDA | p-value* |
| | N(%) | N(%) | N(%) | |
| TOXICITY | | | | 0.52 |
| Pain grade II | 1 (4) | 0 | 2 (1.8) | |
| Leucorrhea | 0 | 2 (1.8) | 3 (2.7) | |
| None | 24 (96) | 109 (98.2) | 105 (95.5) | |

\* p-value comparisons across treatment groups for categorical variables based on the Fisher's exact test.

The toxicity during phase II was observed in only one patient (4%) with pain grade > 7, who required intravenous analgesia on the 23rd postoperative day. It was resolved with a single dose of intravenous analgesia without the need to remove the device. No other cases of toxicity were observed (Table 2). Three devices were removed before the expected date during Phase II (3/25 = 12%). There were two loose devices at 15 days of follow-up that were removed on the 15th postoperative day, one due to a fever, which was resolved with antipyretic alone within 48 hours, and the other after a gynecologic examination. Another device was loose on the 30th postoperative day and was removed.

During phase III, in the first 30 days after surgery, the results of 221 patients from the 240 randomized ones were analyzed, with 111 patients (50.2%) in the DUDA group and 110 (49.8%) in the No DUDA group. A total of 19 patients were excluded from the surgical analysis (after conization surgery): 16 presented with invasive squamous cell carcinoma on final histology examination, and 3 were lost to follow-up. For the 3-month analysis, 202 patients were evaluated, 199 patients at the 6-month evaluation, and 192 patients at the 12-month evaluation (Fig 3). All primary and secondary outcomes were evaluated in all phase III patients. The demographic characteristics of phase III patients showed a homogeneous group (Table 3).

Regarding the final histology examination, we observed 142 cases of CIN III (64.3%), 58 cases of cervicitis (26.2%), 20 cases of CIN II (9%), and 1 case of adenocarcinoma in situ (0.5%). We had four pregnancies during follow-up, one in the DUDA group and three in the No DUDA group (p = 0.36) (Table 1). The patient chose the anesthesia for the procedure in phase III: local in 106 patients (48%) and sedation plus local in 115 patients (52%). The principal investigator of the study performed all surgeries. The median duration of surgery was 15 min (IQR = 6) for the DUDA group and 10 min (IQR = 5) for the No DUDA group (p<0.001) (Table 1).

There were 7 cases of toxicity, 2 in the DUDA group (1.8%) and 5 in the No DUDA group (4.5%). In the DUDA group, we had two cases of grade 1 leukorrhea that were treated clinically without antibiotics. No grade 3 or 4 toxicity was detected. In the No DUDA group, we had two cases of grade II pelvic pain that were clinically resolved, and three cases of grade 1 leukorrhea all resolved clinically without antibiotics. No grade 3 or 4 toxicity was detected. Although there was no statistically significant difference between the groups (p = 0.52), the No DUDA group had a 2.5x higher toxicity rate than the DUDA group (Table 2).

During the evaluation at 30 postoperative days in phase III, three devices (2.7%) had loosened. The first case loosed 19 days after surgery and was removed, and the other two were removed on the 30th postoperative day because they were inside of the vagina, not attached to the cervix. The stenosis was evaluated over time, and there was no statistically significant difference, ranging from 4 to 7.3% and 5.8 to 4.3% in the groups with and without DUDA, respectively (Table 4).

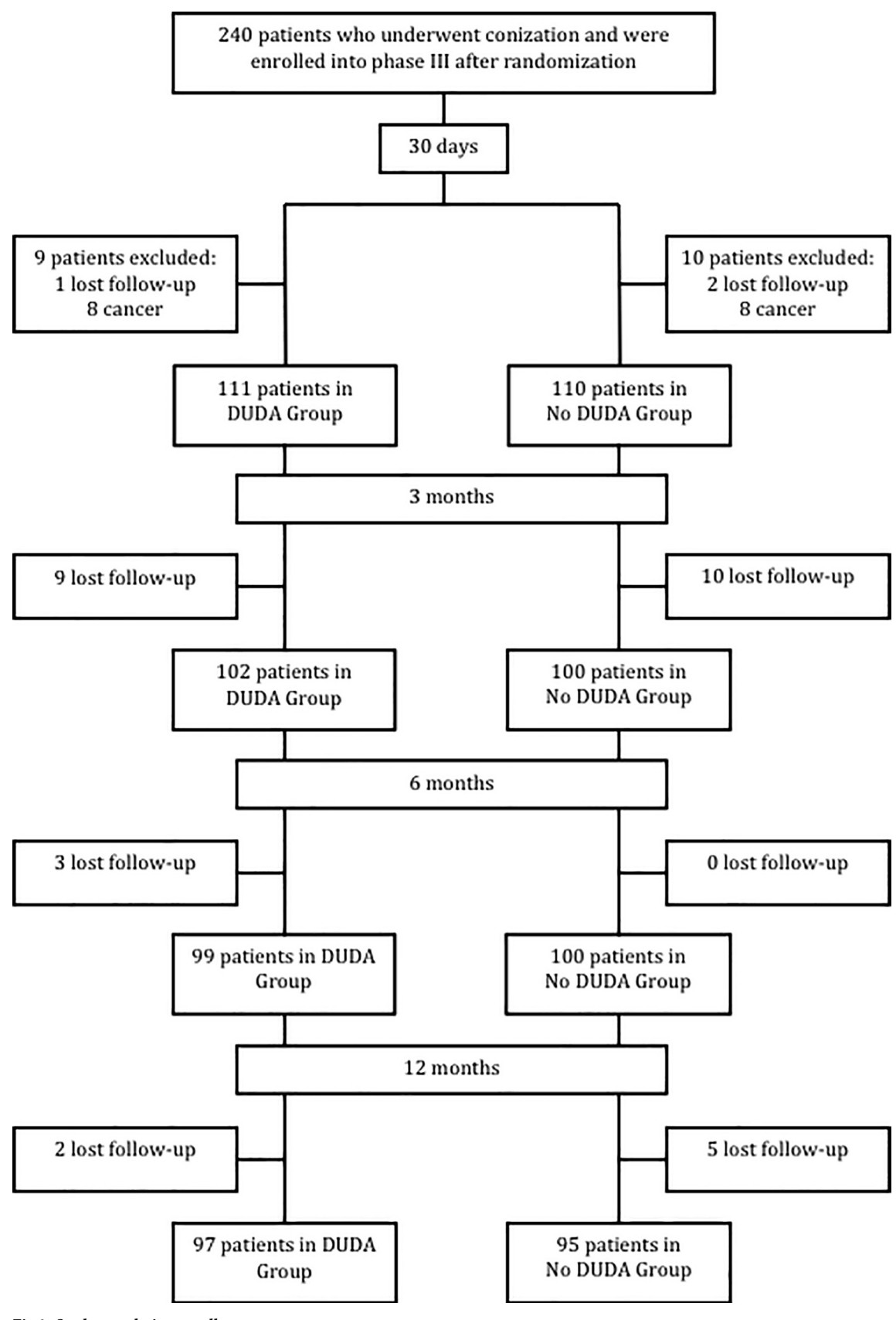

**Fig 3. Study population enrollment.**

**Table 3. Demographic characteristics—phases II and III.**

| | | Phase II | Phase III | Phase III |
|---|---|---|---|---|
| | | N = 25 | DUDA | No DUDA |
| | | | N = 111 | N = 110 |
| *Age (median)* | | 35.16 (22–68) | 34 (21–64) | 34 (23–63) |
| *Race* n (%) | White | 15 (60) | 75 (68) | 75 (68) |
| | Black | 2 (8) | 9 (8) | 6 (6) |
| | Mixed | 8 (32) | 27 (24) | 29 (26) |
| *Pap smear* n (%) | Minor findings | 1 (4) | 5 (9) | 4 (4) |
| | Major findings | 13 (52) | 45 (41) | 44 (40) |
| | Negative/Benign | 11 (44) | 61 (55) | 52 (57) |
| *Colposcopy* n (%) | Negative/Benign | 0 | 3 (3) | 0 |
| | Minor findings | 2 (8) | 2 (2) | 28 (26) |
| | Major findings | 23 (92) | 75 (68) | 76 (69) |
| | Suggestive of invasion | 0 | 0 | 0 |
| *SCJ visualization* n (%) | Visible | 23 (92) | 104 (94) | 105 (96) |
| | Partially visible | 2 (8) | 0 | 0 |
| | Nonvisible | 0 | 7 (6) | 5 (5) |
| *Cervical biopsy* n (%) | CIN II[c] | 8 (32) | 19 (17) | 27 (25) |
| | CIN III[d] | 11 (44) | 58 (52) | 54 (49) |
| | HILs | 6 (24) | 34 (31) | 29 (26) |
| *Intraoperative complications* n (%) | Yes | 0 | 0 | 0 |
| | No | 25 (100) | 111 (100) | 110 (100) |

Normality was assessed using the Kolmogorov-Smirnov normality test, and when necessary, a parametric test (t-test), with a description of the mean (m) and standard deviation (SD), and nonparametric test (Mann-Whitney), with a description of the median (md) and inter-quartile range (IQR), were used.

a- Median (min = minimum, max = maximum)

b–Squamocolumnar junction

c- Cervical Intraepithelial neoplasia grade 2

d- Cervical Intraepithelial neoplasia grade 3.

When comparing the evolution at 3, 6, and 12 months in terms of cervical patency (stenosis after insertion of the hysterometer) and visualization of the squamocolumnar junction during colposcopy, we did not find a significant difference. The DUDA group exhibited 14.1% to 19.4% nonvisualization of the SCJ, whereas that rate ranged from 10.8 to 11.7% in the No DUDA group (Table 4).

**Table 4. Cervical patency—cervical stenosis (%) and SCJ visualization (%).**

| | DUDA | NO DUDA | p-value* |
|---|---|---|---|
| **Cervical stenosis** | N (%) | N (%) | |
| 3 months (N = 202) | 4 (4) | 6 (5.8) | 0.5 |
| 6 months (N = 199) | 5 (5.1) | 9 (9.1) | 0.4 |
| 12 months (N = 192) | 7 (7.3) | 4 (4.3) | 0.5 |
| **Nonvisualization of SCJ[a] (TZ3)[b]** | N (%) | N (%) | |
| 3 months (N = 202) | 14 (14.1) | 11 (10.8) | 0.45 |
| 6 months (N = 199) | 17 (17.3) | 12 (11.9) | 0.5 |
| 12 months (N = 192) | 19 (19.4) | 11 (11.7) | 0.2 |

*- P-value comparisons across treatment groups for categorical variables based on the chi-squared test.

a–Squamocolumnar junction

b- TZ3—transformation zone type 3 (unsatisfactory colposcopy).

## Discussion

Even though this results in our clinical trial showed no difference regarding the anti–cervical stenosis potential after LEEP with the placement of DUDA, in the group in which the device was not placed, there was an evolution over time from a totally visible SCJ to a partially visible SCJ. This finding was owed to the greater difficulty of colposcopic follow-up and thus may be related to greater difficulty for early detection of high-grade lesion recurrence.

Regarding research implications, although the findings of nonrandomized studies are favorable toward the use of an anti–cervical stenosis device, there is still a lack of robust evidence on the safety and efficacy of these methods. It is necessary to explore the application of devices in the cervical canal with greater scientific rigor to prevent cervical stenosis. Studies have shown a high risk of cervical stenosis after cervical conization when there is a larger excision of the entire endocervix (RR 5.07) and when the excision of the cervical canal is > 2 cm (RR 1.95) [14]. It is important to emphasize that excisional procedures allow a detailed evaluation of the entire excised tissue, reducing the risk of inadvertently treating a microinvasive or occult invasive carcinoma as a preinvasive lesion. When the colposcopic evaluation is unsatisfactory (nonvisible squamocolumnar junction), there is an up to 7% chance of a diagnosis of occult invasive carcinoma in the conization specimen by biopsy showing CIN II/III. Thus, diagnostic conization procedures that allow histological examination of the endocervix are usually employed for women with biopsies confirming CIN II/III who have unsatisfactory colposcopic evaluation.

While many studies report cervical stenosis after conization, few studies have described techniques for treating the endocervical canal's stenosis. Those that described some type of intervention were not randomized and had few patients. In addition, they used different types of devices to prevent cervical stenosis. A significant strength of this study is the randomization of the treatment and it is the first prospective randomized clinical trial in a large sample of patients evaluating the antistenosis role of an endocervical device. In our study, we observed a maximum cervical stenosis rate of 8% in the DUDA group and 9% in the No DUDA group. Luesley et al., based on their initial experience with 40% cervical stenosis after conization, conducted a pilot study in 1990 with 33 patients who underwent cold knife conization and placement of an antistenosis device that was secured in the cervix and maintained for 14 days. None of the patients exhibited discomfort during the removal of the device. The second evaluation, in this study, of patency of the endocervical canal was performed on the 6th month, and stenosis was observed in 6% of the DUDA group, though the transformation zone was not fully visible in 36% of them. In contrast to the cold knife conization done by Luesley at el., our study performed LEEP, which has been shown to have lower stenosis rates [22] Given the stenosis and SCJ visualization rates found in our study, considering a power of 80% and a significance level of 5%, the required sample sizes would be at least 3194 patients to find an anti-stenosis difference between the groups.

Studies with anti-stenosis devices have been done in an attempt to reduce cervical stenosis rates. Nasu K and Narahara H reported on four patients subjected to cold knife conization with intrauterine device placement. In the reassessment at 6 months, no recurrence of stenosis was observed in any case [23]. Motegi E et al. reported on two cases and found that the IUD was effective for stenosis after an unsuccessful attempt at endocervical canal dilation alone [29]. This study is interesting because it shows that stenosis can occur after 6 months of manipulation of the cervix, so we analyzed it 6 months after conization. Dilation alone did not solve the problem of cervical stenosis, since stenosis relapse occurred early and frequently. Grund et al. described a case report using a vascular stent that was secured in the cervix after dilation and left for 9 months after laser conization. After removal, the patient became pregnant and

had a successful pregnancy, demonstrating the efficacy of the device in maintaining patency of the endocervical canal [24]. The choice of material to be used in the study was due it is unique and pure (no additives), with excellent dimensional stability, low coefficient of friction, low moisture absorption, and high resistance to abrasion. It maintains its characteristics when immersed in hot water. It exhibits high durability, is resistant to biological attacks and impacts, and can even hold weight for an indefinite period of time. It also has low weight, high melting point, good impact strength, high lubricant ability, thermal insulation, low coefficient of friction, and high resistance to wear and chemical agents, so it is superior to bronze, brass, aluminum, cast iron, and steel for our purposes. In addition to the advantages above, this material is inexpensive. It has a nonporous surface, which prevents 380 the accumulation of substances. consequently, it does not form a biofilm, which reduces the risk of infections associated with it. For the above properties, along with the fact that polyacetal is used in the manufacturing of disposable surgical materials and has very low allergenicity, we choose this material to make the device.

For developing countries or those with low socioeconomic levels, these alternatives are not feasible due to the high cost of the devices and the instruments used for the laser procedure. Our study proposes the use of a device developed specifically to avoid the cervix's stenosis, which is fully anatomically adapted to the size of the endocervical canal and the ectocervix. Besides that, it has low toxicity, safety, and low cost. Another alternative already described in the literature by Yasmin Tan et al. is the use of a urinary catheter stent without a cuff on the distal end. It was evaluated in a case report with five patients to describe the safety, ease of use, and effectiveness of this device for cervical stenosis. That study did not show relapse after use, and of the five patients in the study, four of them had a successful pregnancy without presenting adverse effects [25]. In phase 1 of our study, we had six pregnancies, and during phase 2, we had one pregnancy in the DUDA group and three in the No DUDA group. Although the No DUDA group had a 3x higher rate of pregnancies (2.7% x 0.9%), this difference was not statistically significant. We propose a simple placement technique for a single-use device that can be removed in the doctor's office without anesthesia.

The device's loss rate (2.7%) or infection (4.5%) related to the device was very low, which does not make its use unfeasible. There are reports of the use of devices that maintain cervical patency for a short period time to avoid re-stenosis of the cervical canal and maintain patient fertility or to facilitate follow-up in women of non-fertile age [22–24].

Thus study has limitations. The fact that the surgery was performed by the same surgeon and all patient follow-ups were performed by a single physician could have generated a bias in the study. However, the colposcopic analysis was performed by a second researcher who was blinded to the groups, which provides greater credibility to the data. We believe that although this study did not present results outwardly favorable to the use of an anti–cervical stenosis device, we found no significant toxicity from the device secured in the cervix and kept there for up to 30 days. For this, perhaps a multicenter study could be useful, to combine a larger number of cases in a short period of time, and the final evaluation of the stenosis could be performed with a longer follow-up period to assess whether there is also a long-term impact on cervical stenosis. Another question that still needs to be answered is whether the use of DUDA would change the stenosis rates in laser and cold knife conization. New studies with more patients are needed because cervical stenosis is widespread and there is nothing that considerably decreases its rate.

## Conclusions

This randomized clinical trial that evaluated the role of a new device to prevent cervical stenosis after LEEP conization demonstrated the absence of severe adverse effects that would

preclude placement of the device. There was no statistically significant difference between the use or non-use of the anti-stenosis device to prevent cervical stenosis. Studies with more patients and with standardization of the type of conization to be performed are necessary to draw more robust conclusions. DUDA's use opens up a discussion of cervical stenosis related to the type of conization performed and the use of anti-stenosis devices to lower the stenosis rate.

## Supporting information

**S1 Checklist. CONSORT 2010 checklist.**
(DOC)

**S1 File. CONSORT 2010 flow diagram.**
(DOC)

**S2 File. Study protocol.**
(DOCX)

## Acknowledgments

The authors will like to thank the Barretos Cancer Hospital (BCH), São Paulo, SP, Brazil, Department of Pathology, Prevention and all team from the Teaching and Resource Institute from BCH.

## Author Contributions

**Conceptualization:** Marcelo de Andrade Vieira.

**Data curation:** Marcelo de Andrade Vieira.

**Formal analysis:** Marcelo de Andrade Vieira, Raphael Leonardo Cunha de Araújo, Carlos Eduardo Mattos da Cunha Andrade, Agnaldo Lopes Filho, Ricardo dos Reis.

**Investigation:** Marcelo de Andrade Vieira, Ricardo dos Reis.

**Methodology:** Marcelo de Andrade Vieira, Raphael Leonardo Cunha de Araújo, Carlos Eduardo Mattos da Cunha Andrade, Ronaldo Luis Schmidt, Agnaldo Lopes Filho, Ricardo dos Reis.

**Project administration:** Marcelo de Andrade Vieira, Ricardo dos Reis.

**Resources:** Marcelo de Andrade Vieira, Raphael Leonardo Cunha de Araújo, Ronaldo Luis Schmidt, Agnaldo Lopes Filho, Ricardo dos Reis.

**Supervision:** Ricardo dos Reis.

**Visualization:** Carlos Eduardo Mattos da Cunha Andrade.

**Writing – original draft:** Marcelo de Andrade Vieira, Ricardo dos Reis.

**Writing – review & editing:** Marcelo de Andrade Vieira, Raphael Leonardo Cunha de Araújo, Carlos Eduardo Mattos da Cunha Andrade, Ronaldo Luis Schmidt, Agnaldo Lopes Filho, Ricardo dos Reis.

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
