## [Decision Letter · Decision Letter 0]

7 Apr 2020

PONE-D-20-04004

Randomized Clinical Trial of a new Anti–Cervical Stenosis Device after Conization by Loop Electrosurgical Excision

PLOS ONE

Dear MD vieira,

Thank you for submitting your manuscript to PLOS ONE. After careful consideration, we feel that it has merit but does not fully meet PLOS ONE’s publication criteria as it currently stands. Therefore, we invite you to submit a revised version of the manuscript that addresses the points raised during the review process.

Specifically, the reviewers raised overlapping concerns with the reporting of the definitions, demographic information and methodology of the Clinical Trial protocol. In addition, one reviewer raised concerns about the statistical reporting in the manuscript. 

We would appreciate receiving your revised manuscript by May 21 2020 11:59PM. To enhance the reproducibility of your results, we recommend that if applicable you deposit your laboratory protocols in protocols.io, where a protocol can be assigned its own identifier (DOI) such that it can be cited independently in the future. For instructions see: http://journals.plos.org/plosone/s/submission-guidelines#loc-laboratory-protocols

We look forward to receiving your revised manuscript.

Kind regards,

Richard Hodge

Associate Editor

PLOS ONE

Journal Requirements:

2. Thank you for your ethics statement: 'The study was conducted at the Barretos Cancer Hospital from August 2015 to June 2018. It was started after approval by the hospital and national ethics committees'.

(a)Please amend your current ethics statement to include the full name of the ethics committee/institutional review board(s) that approved your specific study.

(b) Once you have amended this/these statement(s) in the Methods section of the manuscript, please add the same text to the “Ethics Statement” field of the submission form (via “Edit Submission”).

For additional information about PLOS ONE ethical requirements for human subjects research, please refer to " ext-link-type="uri" xlink:type="simple">http://journals.plos.org/plosone/s/submission-guidelines#loc-human-subjects-research."

3. In your Methods section, please provide additional information about the participant recruitment method and the demographic details of your participants. Please ensure you have provided sufficient details to replicate the analyses such as: a) the recruitment date range (month and year) and b) a description of how participants were recruited."

4. During our internal evaluation, the on-house editorial staff noted that the image presented in Figure 3 may be unsuitable for publication. At this time, we ask that you remove it or replace it with a suitable diagram instead. Thank you for your attention to this request.

5. Thank you for stating the following financial disclosure: 'NO'

Reviewers' comments:

Reviewer's Responses to Questions

**Comments to the Author**

1. Is the manuscript technically sound, and do the data support the conclusions?

Reviewer #1: No

Reviewer #2: Yes

Reviewer #3: Yes

2. Has the statistical analysis been performed appropriately and rigorously? 

Reviewer #1: Yes

Reviewer #2: I Don't Know

Reviewer #3: Yes

3. Have the authors made all data underlying the findings in their manuscript fully available?

Reviewer #1: Yes

Reviewer #2: No

Reviewer #3: Yes

4. Is the manuscript presented in an intelligible fashion and written in standard English?

Reviewer #1: No

Reviewer #2: Yes

Reviewer #3: Yes

5. Review Comments to the Author

Reviewer #1: The paper is on an interesting subject: prevention of cervical stenosis following conization.

But I would not recommend this paper for publication in its current form as there are too many problems with this study.

. The main problem is that the primary objective of the study is the prevention of cervical stenosis and the authors don?t give any precision how stenosis is evaluated. Is it with visual inspection or colposcopy ? Is it when the all junction zone is not seen ? Is it when the patency of the os is not obtained with heggar dilatator...?

. Also the way the study was realized in 2 phases is not well defined. And in my opinion it is a phase 2 and a phase 3 study (and not phase 1 and 2).

. The way, the device used should prevent stenosis is not clearly described. Is it mechanical or medical (product used for the device...). Also we have some doubt concerning the pre-study evaluation of the device. Does it has been tested in animal models ? Do we kmow if there are no potential dangerous side effects of the device ?

. Last, an english revision should be performed on the text and mainly the abstract.

Reviewer #2: This study sets out to evaluate the safety of DUDA in a Phase 1 study. Followed by evaluation of efficacy and safety in Phase II study.

The manuscript reads well, however could benefit from a few suggestions below.

1) Under materials and methods- there is the first mention of randomisation process which is in relation to presumably phase II, however randomisation process is also mentioned again under study design Better to mention in one place. Also, although randomisation was done by an online computer program, more detail is required, e.g simple randomisation or was there some element of block randomisation.

2) It’s understood that patients and physicians were not blinded to the allocation group. It would be good to know if there was allocation concealment for those that were recruiting and randomising patients.

3) Also was the statistical analysis carried in unblended or blinded fashion?

4) Authors should explicitly state reason for choosing a sample size of 25 in Phase I. For phase II sample size calculation, range between 104 to 203 per group and in the end 120 per groups were recruited, was this the max number reached in the recruitment period or was this decided as priory since it was in the range of sample size. Was attrition rate into account?

5) The sentence re sample size under statistical analysis (line 101) should really be moved to line 100.

6) Also the sentence in 143 re the blinded gynaecologist, this could to move earlier section in manuscript (i.e. after line 124).

7) According to ICH E9 , its suggested to define populations included in analyses.

8) Recruitment only mentions people randomised, did the authors collect information relating to number screened (also people excluded pre-randomisation), if so it would be good include this information as well.

9) Figure 1 shows all randomised including Phase 1. In my opinion the CONSORT should be focused on Phase II study. Also there should be another level (110 vs 110), followed by the n=19 excluded (indicating number in each group).

10) Table 1, its recommended not to carry out statistical tests at baseline and also Table 1 should only include variables measured at baseline. For an example duration of surgery and pregnancy should not be Table 1.

11) Table 2- is this fishers exact test as opposed to chi-squared and also the numbers are so few to carry formal statistical tests, and chi2 test not replicable, please check –value.

12) Line 247 mentioning range 5 – 8%, please check rounded numbers as table 3 shows 4 – 7%. Same comment for line 252, table 3 shows 14 – 19% and 11- 12%.

13) Again statistical tests in Table 3, ideally should be fishers exact test due to small numbers.

14) Result for physical examinations are not presented as part of the results although these data were collected.

Reviewer #3: Review PONE-D-20-04004:

Randomized Clinical Trial of a new Anti-Cervical Stenosis Device after Conization by Loop Electrosurgical Excision.

Dear Editor,

thank you for the opportunity to read above mentioned manuscript in advance. With pleasure I’ve read this randomized trial. The question asked is relevant since especially in postmenopausal patients cervical stenosis post conization is a problem. The device used was not beneficial. Thus, this is a study with negative findings.

Studies with negative findings have the same importance for the medical community compared to studies with positive findings

However, I do have considerable questions and remarks with respect to current version of the manuscript.

1) The definition of cervical stenosis is not clear and possibly wrong: non-visualization of the SCJ does not imply cervical stenosis! Non-passage of a hysterometer does not imply cervical stenosis!!! A hysterometer usually has a diameter of Hegar 2 or 3, which is not cervical stenosis.

Cervical stenosis post conization is either dysmenorrhea associated with hematometra, complete occlusion of the cervical os on colposcopy or impossibility of taking a follow-up cervical smear using a cytobrush. This was not measured in the study.

2) In addition development of stenosis is highly dependent on menstruation or non-menstruation: if regular menses is present stenosis is rare since blood flow keeps the canal open, whereas in menopause the incidence of stenosis is as high as 20%.

We do not get information how many patients were postmenopausal which is the most important demographic parameter.

3) There is no information about depth of the conisation or volumetric measurement of resected volume, a fact that can directly attribute to later development of cervical stenosis.

4) Additional comments: Line 56 to 70 should be skipped since common knowledge and do not adhere to the topic of stenosis

5) How can the gynecologic be blinded if the patient knows if she had DUDA or not: "The gynecologist who performed the colposcopy was blinded to the groups."

6) Was there any PID between surgery and post day 30 (one patient developed fever, why?)

7) Why did 29 patients not come to all follow-up visits? Could this be explained by morbidity associated with the DUDA device?

8) Why general and local anaesthesia and not either one?

In conclusion, it is a huge merit of Brazilian colleagues having designed and finished this interesting trail. Due to its negative result proposed anti-stenosis device probably will not be implemented in gynecologic practice. Therefore the weaknesses of the trial carry no so much weight in interpretation. Thank you for considering me as a reviewer of this manuscript.

6. PLOS authors have the option to publish the peer review history of their article (what does this mean?). If published, this will include your full peer review and any attached files.

Reviewer #1: Yes: Patrice Mathevet

Reviewer #2: No

Reviewer #3: No

---

## [Author Response · Author response to Decision Letter 0]

2 Jun 2020

Response to Reviewers

PONE-D-20-04004

Reviewer #1

The paper is on an interesting subject: prevention of cervical stenosis following conization.

But I would not recommend this paper for publication in its current form as there are too many problems with this study.

Comment:

1. The main problem is that the primary objective of the study is the prevention of cervical stenosis and the authors don’t give any precision how stenosis is evaluated. Is it with visual inspection or colposcopy ? Is it when the all junction zone is not seen? Is it when the patency of the os is not obtained with heggar dilatator...?

Response

We provide the information requested by the Reviewer in the Follow Up

During the physical examination, the external visualization of the external cervical orifice (visible or nonvisible/stenosis), menstrual flow (days), menstrual cramps after surgery (mild: spontaneous improvement; moderate: improvement after the use of oral medications; and severe: requiring intravenous analgesia) and patency of the endocervical canal were assessed through inserting the hysterometer through the canal, classifying the resistance to insertion as none, little or moderate, or failed insertion (complete resistance to the insertion of the hysterometer). 

Comment:

2. Also the way the study was realized in 2 phases is not well defined. And in my opinion, it is a phase 2 and a phase 3 study (and not phase 1 and 2).

Response

We agree with your comment. The Phase II evaluated the toxicity and safety and the Phase III was done to investigate the efficacy of device compared to standard treatment. We have made the suggested changes. The sentence now reads in Study Design as follows:

We performed phases II and III on the new cervical device according to the standards of randomized clinical trials. 

Comment:

3. The way, the device used should prevent stenosis is not clearly described. Is it mechanical or medical (product used for the device...). Also we have some doubt concerning the pre-study evaluation of the device. Does it has been tested in animal models? Do we know if there are no potential dangerous side effects of the device?

Response

We have provided clarification it. The following has been added to the Materials and Methods / Characteristics of device and sentence now reads as follows:

The device remains in the endocervical canal for 30 days and causes a mechanical obstruction to central healing in an attempt to prevent its stenosis. 

 Comment:

4. Last, an english revision should be performed on the text and mainly the abstract.

Response

With regard to “english revision” this manuscript was reviewed and edited by American Journal Experts again.

Reviewer #2

This study sets out to evaluate the safety of DUDA in a Phase 1 study. Followed by evaluation of efficacy and safety in Phase II study.

The manuscript reads well, however could benefit from a few suggestions below.

Comment:

1. Under materials and methods- there is the first mention of randomization process which is in relation to presumably phase II, however randomisation process is also mentioned again under study design Better to mention in one place. Also, although randomisation was done by an online computer program, more detail is required, e.g simple randomisation or was there some element of block randomisation. 

Response

We appreciate the observation and we have made the suggested changes removing the part of randomization from study design and added this information about randomization process. The sentence now reads as follows in Materials and Methods

After this step an online randomization was carried out on the website. A randomization list was generated randomly using the R. statistical software. This list was a sequence with dichotomous values divided into even-sized blocks in order to ensure balance between patients who would or would not use the DUDA device. Subsequently, this list was attached to a project made on the RedCap Platform to guarantee the fidelity and security of data and results. [26]. This randomization was blinded, but due to the nature of the study, the patients and physicians were not blinded to the intervention after the outcome of the randomization. And the patients were allocated to groups with (DUDA Group) or without placement of the DUDA device (No DUDA Group). It was a prospective and randomized manner 1:1. 

Comment:

2. It’s understood that patients and physicians were not blinded to the allocation group. It would be good to know if there was allocation concealment for those that were recruiting and randomising patients.

Response

We have provided clarification in the Materials and Methods. But both were blinded until randomization. Only after the randomization process they knew in which group they were allocated (with or without DUDA). Obviously, the doctor who was going to place the device should know in which arm of the study the patient was allocated. But the entire randomization process was done blindly. Characterizing the allocation as concealment. It was impossible blind patients and doctors after randomization.

The statement in the Materials and Methods reads as follows:

This randomization was blinded, but due to the nature of the study, the patients and physicians were not blinded to the intervention after the outcome of the randomization. And the patients were allocated to groups with (DUDA Group) or without placement of the DUDA device (No DUDA Group). 

Comment:

3. Also was the statistical analysis carried in unblended or blinded fashion?

Response

We added the information requested by the Reviewer in the Materials and Methods.

The sentence now reads as follows:

A randomization list was generated randomly using the R. statistical software. This list was a sequence with dichotomous values divided into even-sized blocks in order to ensure balance between patients who would or would not use the DUDA device. Subsequently, this list was attached to a project made on the RedCap Platform to guarantee the fidelity and security of data and results. [26]. This randomization was blinded, but due to the nature of the study, the patients and physicians were not blinded to the intervention after the outcome of the randomization. And the patients were allocated to groups with (DUDA Group) or without placement of the DUDA device (No DUDA Group). It was a prospective and randomized manner 1:1. 

Comment:

4. Authors should explicitly state reason for choosing a sample size of 25 in Phase I. For phase II sample size calculation, range between 104 to 203 per group and in the end 120 per groups were recruited, was this the max number reached in the recruitment period or was this decided as priory since it was in the range of sample size. Was attrition rate into account?

Response

In the statistical analysis demonstrated in Materials and Methods we reach in the minimum and maximum number of patients that should be allocated in the study (104-203) during phase II, as mentioned by the reviewer. Therefore, we defined a value of 120 for each arm, considering the attrition rate (loss of follow-up or exit from the study) of 20%, so we would reach in the minimum of 104 patients in each arm of the study.

After calculating the sample size for phase II, considering endocervical canal stenosis rates of 1.3 to 19%, according to the literature, we calculated 19% of 240 patients (total number that would be randomized). Therefore, we reached in the value of 45, being divided into two groups (each group around 23 patients with or without the device). Thus, we considered the ideal number of 25 women (maximum number of stenosis that could be found when compared to the literature without the use of any device) for the assessment of toxicity during phase I.

Comment:

5. The sentence re sample size under statistical analysis (line 101) should really be moved to line 100.

Response

We appreciate the observation and we have made the suggested changes. The sentence now reads as follows:

It was a prospective and randomized manner 1:1. The sample estimate ranged from 104 to 203 in each group.

Comment:

6. Also the sentence in 143 re the blinded gynaecologist, this could to move earlier section in manuscript (i.e. after line 124).

Response

We appreciate the observation and we have made the suggested changes. The sentence now reads as follows:

This randomization was blinded, but due to the nature of the study, the patients and physicians were not blinded to the intervention after the outcome of the randomization. The gynecologist who performed the colposcopy during the follow up was blinded to the groups.

Comment:

7. According to ICH E9, its suggested to define populations included in analyses.

8. Recruitment only mentions people randomised, did the authors collect information relating to number screened (also people excluded pre-randomisation), if so it would be good include this information as well.

Response

We agree with your comment. The inclusion criteria of the study were described at Materials and methods. We have added the following statement to the manuscript:

A total of 265 patients were selected to participate in the Phase III of the study. However, 20 patients were excluded from randomization and did not enter in the study (18 not meeting inclusion criteria and 02 declined to participate in the study) as noted in the CONSORT attachment. All samples underwent LEEP conization. 

Comment:

9. Figure 1 shows all randomised including Phase 1. In my opinion the CONSORT should be focused on Phase II study. Also there should be another level (110 vs 110), followed by the n=19 excluded (indicating number in each group).

Response

We appreciate the observation and we have made the suggested changes. The Phase I has been removed from CONSORT as suggested by the Reviewer and the modifications in the Figure 4 was performed. As Figure 3 was excluded by request of PlosOne, we renamed figure 4 now for Figure 3. This modification can be seen in the in the figure 3 that it was modified according to your suggestions.

Comment:

10. Table 1, its recommended not to carry out statistical tests at baseline and also Table 1 should only include variables measured at baseline. For an example duration of surgery and pregnancy should not be Table 1.

Response

We agree and we have made the suggested changes. The Table 1 was divided in two Tables now. Table 1 with variable measured at baseline and Table 2 with duration of the surgery and pregnancy.

Comment:

11. Table 2- is this fishers exact test as opposed to chi-squared and also the numbers are so few to carry formal statistical tests, and chi2 test not replicable, please check –value.

Response

We agree with the Reviewer and we have made the suggested changes in Table 2.

Comment:

12. Line 247 mentioning range 5 – 8%, please check rounded numbers as table 3 shows 4 – 7%. Same comment for line 252, table 3 shows 14 – 19% and 11- 12%.

Response

The sentence has been corrected and now reads as follows:

The stenosis was evaluated over time, and there was no statistically significant difference, ranging from 4 to 7.3% and 5.8 to 4.3% in the groups with and without DUDA, respectively (Table 4).

The DUDA group exhibited 14.1% to 19.4% nonvisualization of the SCJ, whereas that rate ranged from 10.8 to 11.7% in the No DUDA group (Table 4).

Comment:

13. Again statistical tests in Table 3, ideally should be fishers exact test due to small numbers.

Response

We agree with the Reviewer and we have made the suggested changes in Table 3.

Comment:

14. Result for physical examinations are not presented as part of the results although these data were collected.

Response

We appreciate the observation regarding the physical examinations however during the gynecological physical examination, as described in the study, it was performed in all follow-up returns as subjective variables. These variables were 

categorized to facilitate the interpretation and described in the tables as toxicity (Table 3): degree of pain, leukorrhea, proportion of endocervical stenosis through hysterometry and visualization of SCJ during the colposcopy exam (Table 4). We believe that categorized analysis reduces the chance of bias in the study.

Reviewer #3

Randomized Clinical Trial of a new Anti-Cervical Stenosis Device after Conization by Loop Electrosurgical Excision.

Dear Editor,

thank you for the opportunity to read above mentioned manuscript in advance. With pleasure I’ve read this randomized trial. The question asked is relevant since especially in postmenopausal patients cervical stenosis post conization is a problem. The device used was not beneficial. Thus, this is a study with negative findings.

Studies with negative findings have the same importance for the medical community compared to studies with positive findings

However, I do have considerable questions and remarks with respect to current version of the manuscript.

Comment:

1. The definition of cervical stenosis is not clear and possibly wrong: non-visualization of the SCJ does not imply cervical stenosis! Non-passage of a hysterometer does not imply cervical stenosis!!! A hysterometer usually has a diameter of Hegar 2 or 3, which is not cervical stenosis. Cervical stenosis post conization is either dysmenorrhea associated with hematometra, complete occlusion of the cervical os on colposcopy or impossibility of taking a follow-up cervical smear using a cytobrush. This was not measured in the study.

Response

First of all we know that our study had negative findings but as we know, this the first clinical trial evaluating the antistenosis capacity of a device specificaly built to avoid this complication. Other studies have shown different results but with small number of patients and different devices. We appreciate your observation regarding the definition of cervical stenosis. However, the concept reported by the reviewer refers to the degree of complete stenosis. As can be seen by the reviewer, we described the main complications and symptoms of partial or total cervical stenosis. The statement in the Introduction reads as follows:

The complications inherent to conization include vaginal bleeding, cervical stenosis, amenorrhea, dysmenorrhea and deep dyspareunia [12, 13]. Cervical stenosis is the most important complication due to the clinical repercussions, which may range from menstrual cramps to hematometra, infertility, and the impossibility of early detection of relapse and / or recurrence of the premalignant lesion due to the difficulty of follow-up [12 , 14-17].

And, we have observed in our patients an evolution in the degree of stenosis with progressive symptoms until the complete closure of the endocervical canal followed by intense dysmenorrhea and amenorrhea that can take months to occur. The statement in the Introduction reads as follows:

In addition, the diagnosis can be made up to 28 months after conization [14].

For these reasons and also we believe that there are different degrees of cervical stenosis of the endocervical canal. We respectfully request to the Editor and the reviewer to allow the manuscript remains in its current structure.

Comment:

2. In addition development of stenosis is highly dependent on menstruation or non-menstruation: if regular menses is present stenosis is rare since blood flow keeps the canal open, whereas in menopause the incidence of stenosis is as high as 20%. We do not get information how many patients were postmenopausal which is the most important demographic parameter.

Response

We agree with your comment that cervical stenosis development is highly dependent on menstruation or non-menstruation. This is the reason why we have included in our inclusion criterias only patients aged 18 to 65 years. Some points we took into consideration for the elaboration of this trial. For evaluation of menstrual symptoms, we excluded menopausal patients from the analysis according to your comment. It can be seen in the last paragraph of the Statistical Analysis:

For the symptoms of menstruation and dysmenorrhea, we excluded the menopausal women from the analysis.

Another point that must be taken into account is that when we applied statistical tests comparing non-menopausal and menopausal women, no difference was observed between the groups. As the result of the study was negative, we evaluated that the information would only generate confusion without adding data to the study. In Table 1, we can see that menopausal patients were included, evaluating the age variation of the groups that were part of the study. And, all the studies that we are aware of to date have evaluated both age groups together due to the chance of stenosis in both.

Comment:

3. There is no information about depth of the conisation or volumetric measurement of resected volume, a fact that can directly attribute to later development of cervical stenosis.

Response

We agree with this comment. After the first 30 days, in the first follow-up, the size of the specimen of conization in 3 dimensions (height, depth and width) were analyzed and there was no statistically significant difference between the groups. For this reason, we believe that it would be a confusing criterion for the study without adding relevant information.

Comment:

4) Additional comments: Line 56 to 70 should be skipped since common knowledge and do not adhere to the topic of stenosis

Response

We appreciate and agree with this observation regarding the Additional Comments and these lines (56 to 70) has been removed. However, we request that the introduction remains in this form, because although it is information already known to everyone, for medical students and residents of this area, it would be very interesting to keep this information. Otherwise, we are open to exclude this part if the reviewers and editor will keep this suggestion. 

Comment:

5. How can the gynecologic be blinded if the patient knows if she had DUDA or not: "The gynecologist who performed the colposcopy was blinded to the groups."

Response

We have provided the clarification in the Follow up. as follows:

The gynecologist who performed the colposcopy was blinded to the groups.

The colposcopy exam was performed prior to all follow-ups by the same gynecologist in all patients and was not considered an appointment, as there was an appointment only for this exam. Even the gynecologist only had access to the patient's medical record after the examination. As it was a routine exam, he only knew the follow-up of the patient. After the patient was referred to the oncology surgeon (Study PI) who continued the follow-up during the follow-up. Each visit was preceded by a colposcopy on different days, hence the blindness of the Gynecologist who performed the colposcopies.

Comment:

6. Was there any PID between surgery and post day 30 (one patient developed fever, why?)

Response

The patient who had a fever belonged to the group with DUDA and the report of having a fever at home around the 15th day after surgery, however there was no hemodynamic instability, not reported in the discharge and with normal gynecological examination and the DUDA was well positioned. No other symptoms were reported. When in doubt, the device was removed and as there was no identified infectious focus, only anti-pyretic medication was prescribed. As the clinical situation was resolved in 48 hours, the adverse event was ended. Therefore, DIP was not considered because treatment with antibiotic therapy was not necessary.

Comment:

7. Why did 29 patients not come to all follow-up visits? Could this be explained by morbidity associated with the DUDA device?

Response

We work in a tertiary hospital and patients are referred to us from all over Brazil. We are a country, as you know, with a very large territory and the majority ( 95%) of our patients are from other cities and depend on locomotion offered by the government to come to the hospital. Thus, the loss of follow-up during follow-ups is frequent. However, we have a Research Support Center that makes phone calls to inform a new appointment and / or find out the reason for the loss of appointment. We consider this number as low due to the screening carried out by the hospital's ethics and research department.

Comment:

8. Why general and local anaesthesia and not either one?

Response

As a routine protocol of our institution, we perform local anesthesia in the cervix, in 04 points with Lidocaine 2% as described in Surgical Procedure. However, we observed that some patients in our hospital feel very uncomfortable and request general anesthesia to aid in the procedure. As anesthesia does not influence the outcome of the study, we leave it to patients as an option. And, we have the availability of the anesthetic team to perform this procedure. We believe that in some conizations, more deeply, the general anesthesia is necessary because the patient can bleed and without general anesthesia, the control can be difficult. In small conizations as excision of the transformation zone only, we believe that only local anesthesia is suficient.

---

## [Decision Letter · Decision Letter 1]

27 Oct 2020

A Randomized Clinical Trial of a new Anti–Cervical Stenosis Device after Conization by Loop Electrosurgical Excision

PONE-D-20-04004R1

Dear Dr. vieira,

We’re pleased to inform you that your manuscript has been judged scientifically suitable for publication and will be formally accepted for publication once it meets all outstanding technical requirements.

Within one week, you’ll receive an e-mail detailing the required amendments; in addition to these requirements, please amend the Methods to include the dimensions of the DUDA so that others can reproduce this work. When these have been addressed, you’ll receive a formal acceptance letter and your manuscript will be scheduled for publication.

Kind regards,

Emily Chenette

Deputy Editor-in-Chief

PLOS ONE

Additional Editor Comments (optional):

Reviewers' comments:

Reviewer's Responses to Questions

**Comments to the Author**

1. If the authors have adequately addressed your comments raised in a previous round of review and you feel that this manuscript is now acceptable for publication, you may indicate that here to bypass the “Comments to the Author” section, enter your conflict of interest statement in the “Confidential to Editor” section, and submit your "Accept" recommendation.

Reviewer #1: All comments have been addressed

Reviewer #2: All comments have been addressed

2. Is the manuscript technically sound, and do the data support the conclusions?

Reviewer #1: Yes

Reviewer #2: Yes

3. Has the statistical analysis been performed appropriately and rigorously? 

Reviewer #1: Yes

Reviewer #2: Yes

4. Have the authors made all data underlying the findings in their manuscript fully available?

Reviewer #1: Yes

Reviewer #2: Yes

5. Is the manuscript presented in an intelligible fashion and written in standard English?

Reviewer #1: Yes

Reviewer #2: Yes

6. Review Comments to the Author

Reviewer #1: Authors responded clearly to the different problems assessed during the revision. This paper can be published in its current form.

Reviewer #2: (No Response)

7. PLOS authors have the option to publish the peer review history of their article (what does this mean?). If published, this will include your full peer review and any attached files.

Reviewer #1: **Yes: **Patrice Mathevet

Reviewer #2: No

---

## [Editor Report · Acceptance letter]

16 Nov 2020

PONE-D-20-04004R1 

A Randomized clinical trial of a new anti–cervical stenosis device after conization by loopelectrosurgical excision 

Dear Dr. vieira:

I'm pleased to inform you that your manuscript has been deemed suitable for publication in PLOS ONE. Congratulations! Your manuscript is now with our production department. 

Kind regards, 

on behalf of

Dr Emily Chenette 

Staff Editor

PLOS ONE